# Decentralized sketching of low-rank matrices

**Rakshith S Srinivasa** *
Dept. of Electrical and Computer Engineering
Georgia Institute of Technology
Atlanta, GA 30318
`rsrinivasa6@gatech.edu`

**Kiryung Lee**
Dept. of Electrical and Computer Engineering
Ohio State University
Columbus, OH 43210
`lee.8763@osu.edu`

**Marius Junge**
Dept. of Mathematics
University of Illinois-Urbana Champagne
Urbana, IL, 61801
`mjunge@illinois.edu`

**Justin Romberg**
Dept. of Electrical and Computer Engineering
Georgia Institute of Technology
Atlanta, GA 30318
`jrom@ece.gatech.edu`

## Abstract

We address a low-rank matrix recovery problem where each column of a rank-$r$ matrix $X \in \mathbb{R}^{d_1 \times d_2}$ is compressed beyond the point of individual recovery to $\mathbb{R}^L$ with $L \ll d_1$. Leveraging the joint structure among the columns, we propose a method to recover the matrix to within an $\epsilon$ relative error in the Frobenius norm from a total of $O(r(d_1 + d_2) \log^6(d_1 + d_2)/\epsilon^2)$ observations. This guarantee holds uniformly for all incoherent matrices of rank $r$. In our method, we propose to use a novel matrix norm called the *mixed-norm* along with the maximum $\ell_2$-norm of the columns to design a new convex relaxation for low-rank recovery that is tailored to our observation model. We also show that the proposed mixed-norm, the standard nuclear norm, and the max-norm are particular instances of convex regularization of low-rankness via tensor norms. Finally, we provide a scalable ADMM algorithm for the mixed-norm-based method and demonstrate its empirical performance via large-scale simulations.

## 1 Introduction

A fundamental structural model for data is that the data points lie close to an unknown subspace, meaning that the matrix created by concatenating the data vectors has low rank. We address a particular low-rank matrix recovery problem where we wish to recover a set of vectors from a low-dimensional subspace after they have been individually compressed (or "sketched"). More concretely, let $\mathbf{x}_1, \cdots, \mathbf{x}_{d_2}$ be vectors from an unknown $r$-dimensional subspace in $\mathbb{R}^{d_1}$. We observe the vectors indirectly via linear sketches by corresponding sensing matrices $\mathbf{B}_1, \ldots, \mathbf{B}_{d_2} \in \mathbb{R}^{d_1 \times L}$, where $L < d_1$, i.e., the observed measurement vectors are written as

$$\mathbf{y}_i = \mathbf{B}_i^\top \mathbf{x}_i + \mathbf{z}_i, \quad i = 1, \ldots, d_2. \tag{1}$$

Although individual recovery of each vector is ill-posed, it is still possible to recover $\mathbf{x}_1, \ldots, \mathbf{x}_{d_2}$ jointly by leveraging their mutual structure without knowing the underlying subspace a priori. This indeed results in a low-rank matrix recovery problem with a column-wise observation model.

We are motivated mainly by large-scale inference problems where data is collected in a distributed network or in a streaming setting. In both cases, it is desired to compress the data to lower the

communication overhead. In the first scenario, the data is partitioned according to the network structure and each data point must be compressed without accessing the remainders. In the second scenario, memory or computational constraints may limit access to relatively small number of recent data points.

Such compressive and distributive acquisition schemes arise frequently in numerous real-world applications. In next generation high-resolution astronomical imaging systems, an antenna array may be distributed across a wide geographical area to collect data points that have a high dimension but are also heavily correlated (and so belong to a low-dimensional subspace). Compression at the node level relieves the overhead to transmit data to a central processing unit [1]. In scientific computing, it is common to generate large scale simulation data that has redundancies that manifest as low-rank structures. For example, simulations in a fluid dynamic system generate large state vectors that have low-rank dynamics [2]. Our observational model describes a kind of on-the-fly compression, where the states are compressed as the system evolves, resulting in efficient communication and storage.

In each of these applications, if the underlying low-dimensional subspace were known a priori, then the projection onto that subspace could have implemented an optimal distortion-free linear compression. Alternatively if the uncompressed data were available, the standard Principal Component Analysis (PCA) might have been used to discover the subspace. Unfortunately, neither is the case. Therefore we approach the recovery as sketching without knowing the latent subspace a priori. It is also interpreted as a blind compressed sensing problem that recovers the data points and underlying subspace simultaneously from compressed measurements.

The measurement model in (1) is equivalently rewritten as follows: Let $\mathbf{X}_0 \in \mathbb{R}^{d_1 \times d_2}$ be a matrix obtained by concatenating $\mathbf{x}_1, \ldots, \mathbf{x}_{d_1}$. It follows that the rank of $\mathbf{X}_0$ is at most $r$. The entries of $\mathbf{y}_1, \ldots, \mathbf{y}_{d_2}$ then correspond to noisy linear measurements of $\mathbf{X}_0$, i.e., for $l = 1, \ldots, L$ and $i = 1, \ldots, d_2$, the $l^{\text{th}}$ entry of $\mathbf{y}_i$ denoted by $y_{l,i}$ is written as

$$y_{l,i} = \langle \mathbf{A}_{l,i}, \mathbf{X}_0 \rangle + z_{l,i} \quad \text{with} \quad \mathbf{A}_{l,i} = \frac{1}{\sqrt{L}} \mathbf{b}_{l,i} \mathbf{e}_i^\top, \tag{2}$$

where $z_{l,i}$, $\mathbf{b}_{l,i}$, and $\mathbf{e}_i$ respectively denote the $l^{\text{th}}$ entry of $\mathbf{z}_i$, the $l^{\text{th}}$ column of $\mathbf{B}_i$, and the $i^{\text{th}}$ column of the identity matrix of size $d_2$. We propose a convex optimization method to recover $\mathbf{X}_0$ from $\{y_{l,i}\}$ and provide theoretical analysis when $\mathbf{b}_{l,i}$ and $z_{l,i}$ are independent copies of random vectors drawn according to $\mathcal{N}(0, I_{d_1})$ and $\mathcal{N}(0, \sigma^2)$ respectively.

## 1.1 Mixed-norm-based low-rank recovery

Low-rank matrix recovery has been extensively studied (e.g., see [3]). One popular approach is to formulate the recovery as a convex program with various matrix norms such as the nuclear norm [4, 5, 6] and the max norm [7]. As we show in Section 2, these two norms together with the new norm we propose below are all particular instances of a unified perspective of low-rank regularization. We propose a convex relaxation of low-rankness by a matrix norm for the recovery from measurements given in (2).

For a matrix $\mathbf{X}$, the maximum $\ell_2$ column norm is defined as

$$\|\mathbf{X}\|_{1 \to 2} = \max_{j=1 \cdots d_2} \|\mathbf{X} \mathbf{e}_j\|_2, \tag{3}$$

where $\mathbf{e}_j$ is the $j^{\text{th}}$ standard basis vector. This can be interpreted as the operator norm from the vector space $\ell_1^{d_2}$ to that of $\ell_2^{d_1}$. We define the "mixed-norm" of a matrix $\mathbf{X}$ as

$$\|\mathbf{X}\|_{\text{mixed}} = \inf_{\mathbf{U}, \mathbf{V}: \mathbf{U}\mathbf{V}^\top = X} \|\mathbf{U}\|_{\text{F}} \|\mathbf{V}^\top\|_{1 \to 2}. \tag{4}$$

Indeed the above two norms provide a convex relaxation suitable to the observation model in (2) through the interlacing property given in the following lemma, the proof of which is given in the supplementary material.

**Lemma 1** *Let* $\mathbf{X} \in \mathbb{R}^{d_1 \times d_2}$ *satisfy* $\text{rank}(\mathbf{X}) \leq r$. *Then*

$$\|\mathbf{X}\|_{1 \to 2} \leq \|\mathbf{X}\|_{\text{mixed}} \leq \sqrt{r} \|\mathbf{X}\|_{1 \to 2}. \tag{5}$$

By Lemma 1, the set $\kappa(\alpha, R)$ defined by

$$\kappa(\alpha, R) = \{\mathbf{X} \colon \|\mathbf{X}\|_{1 \to 2} \leq \alpha, \|\mathbf{X}\|_{\text{mixed}} \leq R\} \tag{6}$$

contains the set of rank-$r$ matrices with column norms bounded by $\alpha$. We show that the observation model in (2) results in an $\epsilon$-embedding of the set $\kappa(\alpha, R)$ for a total number of measurements $Ld_2 \gtrsim r(d_1 + d_2) \log^6(d_1 + d_2)/\epsilon^2$. We consider an estimate $\widehat{\mathbf{X}}$ of $\mathbf{X}_0$ given by

$$\hat{\mathbf{X}} \in \underset{\mathbf{X} \in \kappa(\alpha, R)}{\operatorname{argmin}} \sum_{l,i} |y_{l,i} - \langle \mathbf{A}_{l,i}, \mathbf{X} \rangle|^2. \tag{7}$$

We have attempted to use the nuclear norm instead of the mixed norm but this approach was not successful with providing a guarantee at a near optimal sample complexity. Furthermore it also demonstrates worse empirical performance compared to our approach, as show in Section 3.

Another appealing property of the mixed-norm is that it can be computed in polynomial time using a semidefinite formulation. This renders our proposed estimator readily implementable using general purpose convex solvers. However, to address scalability, we propose an ADMM based framework. We defer further details on efficient computation to Section 3.

## 1.2 Main result

Our main result, stated in Theorem 1, provides an upper bound on the Frobenius norm of the error between the estimate $\widehat{\mathbf{X}}$ obtained from solving (7) and the ground truth matrix $\mathbf{X}_0$ that holds simultaneously for all matrices $\mathbf{X} \in \kappa(\alpha, R)$ rather than for a fixed arbitrary matrix $\mathbf{X}_0$. En route to proving our guarantee, we indeed show that $\sum_{l,i} \langle \mathbf{A}_{l,i}, \mathbf{X} \rangle^2$ is well concentrated around its expectation $\|\mathbf{X}\|_{\text{F}}^2$ for all $\mathbf{X} \in \kappa(\alpha, R)$ and hence, the measurements results in an embedding of the set $\kappa(\alpha, R)$ into a low dimension.

**Theorem 1** *Let $\kappa(\alpha, R)$ be defined as in* (6). *Suppose that the $\mathbf{b}_{l,i}$ are drawn independently from $\mathcal{N}(\mathbf{0}, \mathbf{I}_{d_1})$, $(z_{i,l})$ are i.i.d. following $\mathcal{N}(0, \sigma^2)$, $d = d_1 + d_2$ and $d_2 \leq Ld_2 \leq d_1 d_2$. Then, for $R \leq \alpha\sqrt{r}$, there exist numerical constants $c_1$, $c_2$ such that the estimate $\widehat{\mathbf{X}}$ satisfies*

$$\frac{\|\widehat{\mathbf{X}} - \mathbf{X}_0\|_{\text{F}}^2}{\|\mathbf{X}_0\|_{\text{F}}^2} \leq c_1 \cdot \frac{\alpha^2}{\|\mathbf{X}_0\|_{\text{F}}^2 / d_2} \cdot \max\left(1, \frac{\sigma\sqrt{L}}{\alpha}\right) \cdot \sqrt{\frac{r(d_1 + d_2) \log^6 d}{Ld_2}} \tag{8}$$

*with probability at least $1 - 2\exp(-c_2 R^2 d/\alpha^2)$ for all $\mathbf{X}_0 \in \kappa(\alpha, R)$.*

There are a few remarks in order:

- The factor $\alpha^2 d_2 / \|\mathbf{X}_0\|_F^2$ is the ratio between the maximum and the average of the squared column $\ell_2$ norm of the ground truth matrix $\mathbf{X}_0$ and represents its degree of *incoherence*. A ratio close to 1 indicates that the columns have similar $\ell_2$-norms and results in a lower sample complexity than when the ratio is much larger than 1. This is similar to the dependence on the relative magnitude of each entry in the max-norm-based estimator [7] and the dependence on incoherence in matrix completion problems.

- The second factor is written as $\max(1, \eta)$ where $\eta = \frac{\sigma\sqrt{L}}{\alpha}$ accounts for the noise level in the measurements. Since we take $L$ measurements per column and the measurement operator is isotropic, $\alpha^2$, is compared against the corresponding noise-variance $\sigma^2 L$.

- If the incoherence term is upper-bounded by a constant and the normalized noise level $\eta$ satisfies $\eta = \Omega(1)$, then $\widehat{\mathbf{X}}$ obtained from $O(\eta^2 rd \log^6(d)\epsilon^{-2})$ measurements satisfies $\|\widehat{\mathbf{X}} - \mathbf{X}_0\|_{\text{F}}^2 \leq \epsilon \|\mathbf{X}\|_{\text{F}}^2$ with high probability.

- We conjecture that the corresponding minimax lower bound coincide with (8) except the maximum of $\eta$ with 1 and the logarithmic term. Particularly if $\eta = \Omega(1)$, then the sample complexity in (8) will be near optimal.

## 1.3 Related work

The model in (2) has been studied in the context of compressed principal components estimation [8, 9, 10]. These works studied a specific method that computes the underlying subspace though an empirical covariance estimation. While being guaranteed at a near optimal sample complexity, this approach is inherently limited to the linear observation model. On the other hand, our method is more flexible in terms of its potential extension to nonlinear observation models.

Negahban and Wainwright [11] considered the multivariate linear regression problem where a similar model to (2) arises but with a fixed sensing matrix $\mathbf{A}$, i.e., $\mathbf{A}_i = \mathbf{A}$ for all $i = 1, \ldots, d_2$. They showed that a nuclear-norm penalized least squares provides robust recovery at a near optimal sample complexity within a logarithmic factor of the degrees of freedom of rank-$r$ matrices. However, their guarantees applies to an arbitrary fixed ground truth matrix and not to all matrices within the model simultaneously. Our aim is to work with an embedding of the model set $\kappa(\alpha, R)$ and we obtain a *uniform* theoretical guarantee over the entire model set at the cost of using different sensing matrices $\mathbf{A}_i$'s and incoherence of the matrices.

Our solution approach is partly inspired by earlier works on low-rank matrix completion using the max-norm [12, 13, 7]. The pair of max-norm and $\ell_\infty$ norms is used to relax the set of low-rank matrices to a convex model. We generalize this approach to that of using tensor norms (see Section 2) as a proxy for low rank regularization and show that the max-norm and the mixed-norm are particular instances of this general framework. In particular we choose a specific pair of tensor norms in accordance with the structure in the observation model. This leads to a new convex relaxation model of low-rankness, a corresponding optimization formulation, algorithm, and its performance guarantee. Finally, we point out that our method of proofs and the technical tools we use to establish our results are significantly different from that of [7].

## 2 Properties of tensor norms on low-rank matrices

We interpret a matrix $\mathbf{X} \in \mathbb{R}^{d_1 \times d_2}$ as a linear operator from a vector space $\mathbb{R}^{d_2}$ to another vector space $\mathbb{R}^{d_1}$. Then let the domain and range spaces be respectively endowed with the $\ell_p$ norm and the $\ell_q$ norm. The vector space of all $d_1 \times d_2$ matrices is then identified as the tensor product of the two Banach spaces, denoted as $\ell_{p'} \otimes \ell_q$ (e.g., [14]), where $1/p + 1/p' = 1$.

A tensor norm is a norm on the algebraic tensor product of two Banach spaces that satisfies the operator ideal property (see e.g., [14, 15]). The main insight driving the unified perspective is that, when we restrict linear operators to those of rank at most $r$, certain tensor norms become equivalent up to a function of $r$. In particularly, we consider the *injective* and *projective* tensor norms, defined respectively as

$$\|\mathbf{X}\|_\vee = \sup_{\mathbf{u} \in \mathbb{R}^{d_1}, \|\mathbf{u}\|_p = 1} \|\mathbf{X}\mathbf{u}\|_q \tag{9}$$

and

$$\|\mathbf{X}\|_\wedge = \inf \left\{ \sum_k \|\mathbf{u}_k\|_{p'} \|\mathbf{v}_k\|_q \;\middle|\; \mathbf{X} = \sum_k \mathbf{v}_k \mathbf{u}_k^* \right\}. \tag{10}$$

The pair of the injective and projective norms characterizes the set of low-rank matrices through an interlacing property between them. For example, when $p = q = 2$, it can be easily verified that $\|\mathbf{X}\|_\vee = \|\mathbf{X}\|_2$ and $\|\mathbf{X}\|_\wedge = \|\mathbf{X}\|_*$. It follows from the singular value decomposition that $\|\mathbf{X}\|_2 \leq \|\mathbf{X}\|_* \leq r\|\mathbf{X}\|_2$. In yet another example where $p = 1, q = \infty$, we have $\|\mathbf{X}\|_\vee = \|\mathbf{X}\|_\infty$. Linial et al. [13] showed that Grothendieck's inequality implies

$$\|\mathbf{X}\|_\infty \leq \|\mathbf{X}\|_{\max} \leq \sqrt{r} \|\mathbf{X}\|_\infty, \tag{11}$$

where

$$\|\mathbf{X}\|_{\max} = \inf_{\mathbf{U}, \mathbf{V}: \mathbf{U}\mathbf{V}^\top = X} \left\|\mathbf{U}^\top\right\|_{1 \to 2} \left\|\mathbf{V}^\top\right\|_{1 \to 2}.$$

In this case, it has been shown that the max norm is equivalent up to a constant to the projective norm.

Finally, by letting $p = 1$, $q = 2$, we obtain $\|\mathbf{X}\|_\vee = \|\mathbf{X}\|_{1 \to 2}$ and that the projective norm is equivalent (up to a constant factor) to the mixed norm and the relationship in Lemma 1 holds. Further, it is interesting that unlike many tensor norms, the mixed norm and max-norms can be computed efficiently in a polynomial time, similar to the nuclear norm. As we note in the next section, this enables efficient implementation of mixed-norm-based low-rank recovery programs.

# 3 Fast algorithm for mixed-norm-based optimization

The mixed-norm of any matrix $\mathbf{X}$ can be computed in polynomial time as

$$\|\mathbf{X}\|_{\text{mixed}} = \min_{\mathbf{W}_{11}, \mathbf{W}_{22}} \quad \max(\text{trace}(\mathbf{W}_{11}), \|\text{diag}(\mathbf{W}_{22})\|_\infty)$$
$$\text{s.t.} \quad \begin{bmatrix} \mathbf{W}_{11} & \mathbf{X} \\ \mathbf{X}^\top & \mathbf{W}_{22} \end{bmatrix} \succeq \mathbf{0}, \tag{12}$$

where $\text{diag}(\mathbf{W}_{22})$ denotes the vector of the diagonal entries of $\mathbf{W}_{22}$. Then the optimization routine in (7) can be written as

$$\min_{\mathbf{W}_{11}, \mathbf{W}_{22}, \mathbf{X}} \quad \sum_{l,i} |y_{l,i} - \langle \mathbf{A}_{l,i}, \mathbf{X} \rangle|^2$$
$$\text{subject to} \quad \text{trace}(\mathbf{W}_{11}) \le R, \quad \|\text{diag}(\mathbf{W}_{22})\|_\infty \le R,$$
$$\|\mathbf{X}\|_{1\to 2} \le \alpha, \quad \mathbf{W} = \begin{bmatrix} \mathbf{W}_{11} & \mathbf{X} \\ \mathbf{X}^\top & \mathbf{W}_{22} \end{bmatrix} \succeq \mathbf{0}. \tag{13}$$

The program in (13) is now a constrained convex optimization problem over the cone of positive semidefinite (PSD) matrices.

## 3.1 ADMM based fast algorithm

The program in (13) can be implemented using standard convex optimization solvers like SeDuMi. [16]. However, this could result in scaling issues, as run times could be prohibitive in higher dimensions. To address this, we propose to use the ADMM based algorithm [17] which breaks down the optimization problem into smaller problems that can be solved efficiently. Our approach is similar to [18], where the positive semidefinite constraint on $\mathbf{W}$ in (13) is treated separately from the other constraints. We provide an algorithm for the norm-penalized version of (13). By Lagrangian duality, the penalized version and the constrained version are equivalent when the Lagrangian multipliers $\lambda_1$ and $\lambda_2$ are chosen appropriately.

By introducing an auxiliary variable $\mathbf{T}$, it is straightforward to show that the optimization problem (13) is equivalent to

$$\min_{\mathbf{W}, \mathbf{T}} \quad \sum_{l,i} |y_{l,i} - \langle \mathbf{A}_{l,i}, \mathbf{W} \rangle|^2 + \lambda_1 \text{trace}(\mathbf{T}_{11}) + \lambda_2 \|\text{diag}(\mathbf{W}_{22})\|_\infty$$
$$\text{subject to} \quad \|\mathbf{W}_{12}\|_{1\to 2} \le \alpha, \quad \mathbf{T} = \mathbf{W}, \quad \mathbf{T} \succeq 0. \tag{14}$$

In (14), we carry the constraints on $\text{trace}(\mathbf{T}_{11})$ and $\|\text{diag}(\mathbf{W}_{22})\|$ to the objective function by using the Lagrangian formulation. Note that there are other variations possible, with more or fewer constraints carried over to the objective function. The formulation in (14) is amenable to the ADMM algorithm. The augmented Lagrangian of (14) is given by

$$L(\mathbf{T}, \mathbf{W}, \mathbf{Z}) = f(\mathbf{W}) + \lambda_1 \text{trace}(\mathbf{T}_{11}) + \lambda_2 \|\text{diag}(\mathbf{W}_{22})\|_\infty$$
$$+ \langle \mathbf{Z}, \mathbf{T} - \mathbf{W} \rangle + \frac{\rho}{2} \|\mathbf{T} - \mathbf{W}\|_F^2 + \chi_{\{\mathbf{T} \succeq 0\}} + \chi_{\{\|\mathbf{W}_{12}\|_{1\to 2} \le \alpha\}},$$

where $\mathbf{Z}$ is the dual variable and $\chi_{\mathcal{S}}$ is the indicator function of the set $\mathcal{S}$ given as $\chi_{\mathcal{S}}(t) = 0$ if $t \in \mathcal{S}$ and $\chi_{\mathcal{S}}(t) = \infty$ otherwise. The ADMM algorithm then iterates by alternating among $\mathbf{T}$, $\mathbf{W}$ and $\mathbf{Z}$, as shown in Algorithm 1. While we leave the finer details of the algorithm to the supplementary material, it is worthwhile to note that each step in Algorithm 1 has a unique closed-form solution that allows for scalability to high dimensions.

## 3.2 Experiments

To complement our theoretical results, we observe the empirical performance of the mixed-norm-based method in a set of Monte Carlo simulations. Matrices are set to be of size $1,000 \times 1,000$ and of rank 5. In our experiments we normalize the columns to have the same energy. We observe the estimation error by varying the degree of compression and the signal-to-noise (SNR) ratio. We compare the proposed method to the popular matrix LASSO, which minimizes the least squares loss

---

**Algorithm 1** ADMM algorithm

---

**Initialize: $\mathbf{T}^0, \mathbf{W}^0, \mathbf{Z}^0$**
**while** not converged **do**
$\qquad \mathbf{T}^{k+1} = \underset{\mathbf{T} \succeq 0}{\operatorname{argmin}}\, L(\mathbf{T}, \mathbf{W}^k, \mathbf{Z}^k)$
$\qquad \mathbf{W}^{k+1} = \underset{\|\mathbf{W}_{12}\|_{1\to2}\le\alpha}{\operatorname{argmin}}\, L(\mathbf{T}^{k+1}, \mathbf{W}, \mathbf{Z}^k)$
$\qquad \mathbf{Z}^{k+1} = \mathbf{Z}^k + \rho(\mathbf{T}^{k+1} - \mathbf{W}^{k+1})$
**end while**

---

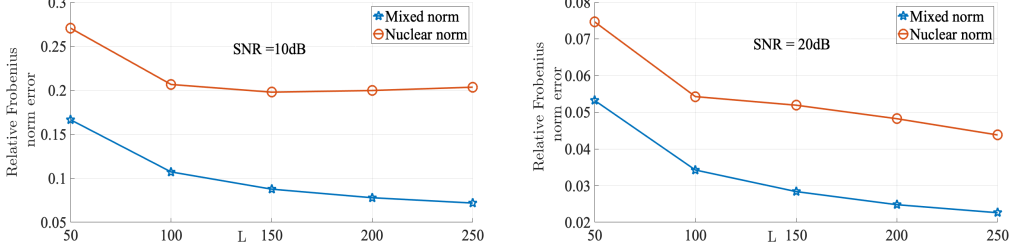

Figure 1: Simulation results comparing the proposed mixed-norm based estimator and the nuclear norm based estimator. The test matrices were of size $1,000 \times 1,000$ with rank 5. Each data point is computed as an average of 5 trials. Mixed norm estimator is able to achieve much lower errors with fewer measurements compared to the nuclear norm estimator.

with a nuclear norm regularizer. We used Algorithm 1 to implement the mixed-norm based method. The nuclear norm minimization approach was implemented using the algorithm provided in [19].

Figure 1 shows the obtained simulation results. The estimation error is averaged over 5 trials. The result indicates that the mixed-norm-based estimator outperforms the nuclear-norm-based estimator at both the SNR levels considered.

# 4 Proof sketch

We state the key lemmas involved in proving our result and point to the tools we use and defer finer details to the supplementary material. We begin with the basic optimality condition that relates the estimate $\hat{\mathbf{X}}$ to the ground truth $\mathbf{X}_0$. Let $\mathbf{M} = \hat{\mathbf{X}} - \mathbf{X}_0$. By the triangular inequality, we have $\mathbf{M} \in \kappa(2\alpha, 2R)$. For notational brevity, we assume from now on that $\mathbf{M} \in \kappa(\alpha, R)$. (Neither the main result nor the proofs are affected by this since they involve multiplication with some numerical constants.)

We adapt the first step in the analysis framework of the analogous matrix completion problem [7]. By optimality of the solution and (2), we have

$$\sum_{l,i} \left( y_{l,i} - \langle \mathbf{A}_{l,i}, \widehat{\mathbf{X}} \rangle \right)^2 \le \sum_{l,i} \left( y_{l,i} - \langle \mathbf{A}_{l,i}, \mathbf{X}_0 \rangle \right)^2 . \tag{15}$$

After substituting $\hat{\mathbf{X}} - \mathbf{X}_0$ by $\mathbf{M}$ and rearranging the terms, we obtain

$$\sum_{l,i} \langle \mathbf{A}_{l,i}, \mathbf{M} \rangle^2 \le 2\sum_{l,i} \langle \mathbf{A}_{l,i}, \mathbf{M} \rangle z_{l,i}. \tag{16}$$

As in [7], we rely on the stochastic nature of the noise. The proof also relies on the norm-constrained optimization rather than norm-penalized optimization. Our strategy is to obtain a lower bound on the quadratic form $\sum_{l,i} \langle \mathbf{A}_{l,i}, \mathbf{M} \rangle^2$ in terms of $\|\mathbf{M}\|_{\mathrm{F}}^2$ and a uniform upper bound on the linear form $\sum_{l,i} \langle \mathbf{A}_{l,i}, \mathbf{M} \rangle z_{l,i}$ over the set $\kappa(\alpha, R)$. We can then bound $\|\mathbf{M}\|_{\mathrm{F}}^2$ uniformly over the set.

## 4.1 Lower bound on the quadratic form

We observe that $\sum_{l,i}\langle \mathbf{A}_{l,i}, \mathbf{M}\rangle^2$ can be reformulated as a quadratic form in standard Gaussian random variables. Let us define

$$\xi = \begin{bmatrix} \mathbf{b}_{1,1} \\ \vdots \\ \mathbf{b}_{L,d_2} \end{bmatrix} \in \mathbb{R}^{Ld_1 d_2}. \tag{17}$$

Then it follows that $\xi \sim \mathcal{N}(0, I_{Ld_1 d_2})$. Therefore, the left-hand side of (16) is rewritten as

$$\sum_{l,i}\langle \mathbf{A}_{l,i}, \mathbf{M}\rangle^2 = \|\mathbf{Q_M}\xi\|^2, \quad \text{where}$$

$$\mathbf{Q_M} = \frac{1}{\sqrt{L}}\begin{bmatrix} \widetilde{\mathbf{M}}_1^\top & 0 & \cdots & 0 \\ 0 & \widetilde{\mathbf{M}}_2^\top & \cdots & 0 \\ \vdots & \vdots & \ddots & 0 \\ 0 & 0 & \cdots & \widetilde{\mathbf{M}}_{d_2}^\top \end{bmatrix}, \quad \widetilde{\mathbf{M}}_j^\top = \mathbf{I}_L \otimes (\mathbf{M}e_j)^\top \in \mathbb{R}^{L \times Ld_1}. \tag{18}$$

We also have

$$\mathrm{E}\,\|\mathbf{Q_M}\xi\|^2 = \|\mathbf{M}\|_{\mathrm{F}}^2$$

We compute a tail estimate on $\sup_{\mathbf{M}\in\kappa(\alpha,R)}\|\mathbf{Q_M}\xi\|_2^2$ by using the results on suprema of chaos processes [20]. They derived a sharp tail estimate on the supremum of a Gaussian quadratic form maximized over a given set $\mathcal{A}$, which is written as

$$\sup_{\mathbf{A}\in\mathcal{A}}\left|\|\mathbf{A}\xi\|^2 - \mathrm{E}\,\|\mathbf{A}\xi\|^2\right|,$$

by using a chaining argument. By adapting their framework, we obtain the following Lemma:

**Lemma 2** *Under the assumptions of Theorem 1, if $\mathbf{Q_M}$ and $\xi$ are as defined in* (18) *and* (17), *then*

$$\sup_{\mathbf{M}\in\kappa(\alpha,R)}\left|\frac{\|\mathbf{Q_M}\xi\|^2}{d_2} - \frac{\|\mathbf{M}\|_{\mathrm{F}}^2}{d_2}\right| \le cR\sqrt{\frac{d}{Ld_2}}\left(\alpha + \frac{R\sqrt{d}}{\sqrt{Ld_2}}\right)\log^3(d)$$

*with probability at least $1 - 2\exp(-cR^2 d\alpha^2)$.*

From Lemma 2, in the regime where $Ld_2 > R^2 d/\alpha^2$, we can obtain

$$\frac{\sum_{l,i}\langle \mathbf{A}_{l,i}, \mathbf{M}\rangle^2}{d_2} \ge \frac{\|\mathbf{M}\|_F^2}{d_2} - cR\alpha\sqrt{\frac{d}{Ld_2}}\log^3 d. \tag{19}$$

## 4.2 Upper bound on the right-hand side of (16)

We obtain the following uniform upper bound $\sum_{l,i}\langle \mathbf{A}_{l,i}, \mathbf{M}\rangle z_{l,i}$:

**Lemma 3** *Under the assumptions of Theorem 1, with probability at least $1 - 2\exp(-cR^2 d/\alpha^2)$,*

$$\sup_{\mathbf{M}\in\kappa(\alpha,R)}\frac{\sum_{l,i}\langle \mathbf{A}_{l,i}, \mathbf{M}\rangle z_{l,i}}{d_2} \le c(\sigma\sqrt{L})R\sqrt{\frac{d}{Ld_2}}\log^3 d. \tag{20}$$

To derive Lemma 3, we first express the left-hand side of (20) using a matrix norm.

Define

$$|||\mathbf{M}||| := \frac{\|\mathbf{M}\|_{1\to 2}}{\alpha} \vee \frac{\|\mathbf{M}\|_{\mathrm{mixed}}}{R}.$$

Then by the definition of $\kappa(\alpha, R)$ in (6) it follows that the unit ball $B := \{\mathbf{M} : |||\mathbf{M}||| \le 1\}$ with respect to $||| \cdot |||$ coincides with $\kappa(\alpha, R)$. Therefore via the Banach space duality, we obtain

$$\sup_{\mathbf{M}\in\kappa(\alpha,R)}\sum_{l,i}\langle \mathbf{A}_{l,i}, \mathbf{M}\rangle z_{l,i} = \sup_{\mathbf{M}\in\kappa(\alpha,R)}\langle\sum_{l,i} z_{l,i}\mathbf{A}_{l,i}, \mathbf{M}\rangle = |||\sum_{l,i} z_{l,i}\mathbf{A}_{l,i}|||_*$$

where $||| \cdot |||_*$ denotes the dual norm. Then, conditioned on $\mathbf{A}_{l,i}$'s, it follows from Theorem 4.7 in [21] that with probability $1 - \delta$

$$\underbrace{||| \sum_{l,i} z_{l,i} \mathbf{A}_{l,i} |||_* \leq \mathrm{E}_z ||| \sum_{l,i} z_{l,i} \mathbf{A}_{l,i} |||_*}_{T_1} + \underbrace{\pi \sqrt{\frac{\log(2/\delta)}{2} \sup_{M \in \kappa(\alpha, R)} \sum_{l,i} \langle \mathbf{A}_{l,i}, \mathbf{M} \rangle^2}}_{T_2} . \quad (21)$$

The first term $T_1$ is the *Gaussian complexity* of the sample set $\{A_{l,i}\}$ over the function class $\{\langle M, \cdot \rangle : M \in \kappa(\alpha, R)\}$. This can be (up to a logarithmic factor of the size of the summation) upper-bounded by the corresponding Rademacher complexity ([22], Equation (4.9)) as

$$T_1 \leq c\sigma \sqrt{\log(Ld_2 + 1)} \, \mathrm{E}_{(r_{l,i})} ||| \sum_{l,i} r_{l,i} \mathbf{A}_{l,i} |||_*, \quad (22)$$

where $(r_{l,i})$ is a Rademacher sequence and the expectation is conditioned on $(A_{i,l})$. Then by the symmetry of the standard Gaussian distribution, we obtain

$$\mathrm{E}_{(r_{l,i})} ||| \sum_{l,i} r_{l,i} \mathbf{A}_{l,i} |||_* = \frac{1}{\sqrt{L}} \sup_{M \in \kappa(\alpha, R)} \left| \sum_{l,i} \langle r_{l,i} \mathbf{b}_{l,i}, \mathbf{M} \mathbf{e}_l \rangle \right| = \frac{1}{\sqrt{L}} \underbrace{\sup_{M \in \kappa(\alpha, R)} \left| \sum_{l,i} \langle \mathbf{b}_{l,i}, \mathbf{M} \mathbf{e}_l \rangle \right|}_{(\S)},$$
$$(23)$$

where the second equation holds in the sense of distribution.

Note that $(\S)$ is the maximum of linear combinations of Gaussian variables and an upper bound can be obtained using Dudley's inequality [22]. Once we obtain a tail estimate of $(\S)$, since $(\S)$ no longer depends on the Rademacher sequence $(r_{l,i})$, it can be used to upper-bound $T_1$ through (22) and (23). An upper bound on $T_2$ has been already derived in Lemma 2. Combining these upper estimates on $T_1$ and $T_2$ results in Lemma 3. From the lower bound on $\sum_{l,i} \langle \mathbf{A}_{l,i}, \mathbf{M} \rangle^2$, we have

$$\frac{\|\mathbf{M}\|_{\mathrm{F}}^2}{d_2} - c\alpha R \sqrt{\frac{d}{Ld_2}} \log^3 d \leq \frac{1}{d_2} \sum_{l,i} \langle \mathbf{A}_{l,i}, \mathbf{M} \rangle^2 \leq \sup_{M \in \kappa(\alpha, R)} \frac{\sum_{l,i} \langle \mathbf{A}_{l,i}, \mathbf{M} \rangle z_{l,i}}{d_2}.$$

From Lemma 3, we get the following inequality, which then leads to the final result.

$$\frac{\|\mathbf{M}\|_{\mathrm{F}}^2}{d_2} \leq c \log^3 dR \sqrt{\frac{d}{Ld_2}} (\alpha \vee \sigma \sqrt{L})$$

## 4.3 Entropy estimate

Part of proofs of lemmas 2 and 3 has been deferred to the supplementary material. Both proofs rely on a key quantity that captures the "complexity" of the set $\kappa(\alpha, R)$. In particular, using Dudley's inequality requires an estimate of the entropy number of the set $\kappa(\alpha, R)$, which is given by the following Lemma.

**Lemma 4** *Let $\kappa(\alpha, R)$ be as in (6) and let $B_{1 \to 2}$ be the unit ball with respect to $\|\cdot\|_{1 \to 2}$. Then there exists a numerical constant $c$ such that*

$$\int_0^\infty \sqrt{\log N(\kappa(\alpha, R), \eta B_{1 \to 2})} d\eta \leq cR\sqrt{d} \log^{3/2}(d_1 + d_2). \quad (24)$$

Here $N(\kappa(\alpha, R), \eta B_{1 \to 2})$ denotes the covering number of $\kappa(\alpha, R)$ with respect to the scaled unit ball $\eta B_{1 \to 2}$.

In Section 2 we introduced the projective tensor norm $\|\cdot\|_\wedge$. Let $B_\wedge$ denote the unit ball with respect to the projective tensor norm in $\ell_\infty^{d_2} \otimes \ell_2^{d_1}$. The injective tensor norm in $\ell_\infty^{d_2} \otimes \ell_2^{d_1}$ reduces to $\|\cdot\|_{1 \to 2}$. By its construction, $\kappa(\alpha, R)$ is given as the intersection of two norm balls $\alpha B_{1 \to 2}$ and $RB_\wedge$. The proof of Lemma 4 reduces to the computation of the entropy number of the identity map on $\ell_\infty^{d_2} \otimes \ell_2^{d_1}$ from the Banach space with the projective tensor norm to that with the injective tensor norm. This proof along with a study of the machinery of computing such entropy numbers can be found in a complementary paper [23].

## 5 Discussion

Low rank modeling is a widely used approach in many machine learning and signal processing tasks. By interpreting low-rankness as a property expressed by tensor norms, we are able to design a practical and sample efficient regularization method that is tailored to the observation model. The proposed method comes with theoretical guarantees and also performs well empirically. Our proposed method can also be implemented efficiently in high dimensions, making it a viable option for performing PCA or low rank recovery in big data scenarios.

## Footnotes

*This work was supported in part NSF CCF-1718771, NSF DMS 18-00872 and in part by C-BRIC, one of six centers in JUMP, a Semiconductor Research Corporation (SRC) program sponsored by DARPA.

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
