[Supplementary Material]

# Supplementary file - Decentralized sketching of low-rank matrices

## 1 Proof of Lemma 1

Let $(\mathbf{U}, \mathbf{V})$ be a minimizer to (4). Then by the feasibility it satisfies $\mathbf{X} = \mathbf{U}\mathbf{V}^\top$. Therefore

$$
\begin{aligned}
\|\mathbf{X}\|_{1\to 2} &= \left\|\mathbf{U}\mathbf{V}^\top\right\|_{1\to 2} \\
&= \max_i \left\|\mathbf{U}\mathbf{V}^\top \mathbf{e}_i\right\|_2 \le \max_i \|\mathbf{U}\| \left\|\mathbf{V}^\top \mathbf{e}_i\right\|_2 \\
&\le \max_i \|\mathbf{U}\|_{\mathrm{F}} \left\|\mathbf{V}^\top \mathbf{e}_i\right\|_2 \\
&= \|\mathbf{U}\|_{\mathrm{F}} \|\mathbf{V}\|_{2,\infty} = \|\mathbf{X}\|_{\mathrm{mixed}}.
\end{aligned}
$$

Suppose that $(\mathbf{U}, \mathbf{V})$ satisfies $\mathbf{M} = \mathbf{U}\mathbf{V}^\top$ and $\mathbf{U}^\top \mathbf{U} = I_r$. Such $(\mathbf{U}, \mathbf{V})$ always exists. For example, think about the SVD of $\mathbf{M}$. Since $(\mathbf{U}, \mathbf{V})$ is feasible to (3), it follows that

$$
\|\mathbf{X}\|_{\mathrm{mixed}} \le \|\mathbf{U}\|_{\mathrm{F}} \left\|\mathbf{V}^\top\right\|_{1\to 2} \le \sqrt{r} \|\mathbf{U}\| \left\|\mathbf{V}^\top\right\|_{1\to 2} = \sqrt{r} \left\|\mathbf{V}^\top\right\|_{1\to 2}.
$$

On the other hand,

$$
\|\mathbf{X}\|_{1\to 2} = \max_i \left\|\mathbf{U}\mathbf{V}^\top e_i\right\|_2 = \max_i \left\|\mathbf{V}^\top e_i\right\|_2 = \left\|\mathbf{V}^\top\right\|_{1\to 2}.
$$

We have shown

$$
\|\mathbf{X}\|_{\mathrm{mixed}} \le \sqrt{r} \|\mathbf{X}\|_{1\to 2}.
$$

In summary, we have

$$
\|\mathbf{X}\|_{1\to 2} \le \|\mathbf{X}\|_{\mathrm{mixed}} \le \sqrt{r} \|\mathbf{X}\|_{1\to 2}.
$$

That is, the pair of $\|\cdot\|_{\mathrm{mixed}}$ and $\|\cdot\|_{1\to 2}$ can be also used for a surrogate of the rank of a matrix.

## 2 Proof of Lemma 2

We derive a tail estimate on $\sup_{\mathbf{M}} \|\mathbf{Q}_{\mathbf{M}} \xi\|_2^2$ by using the results on suprema of chaos processes [1] summarized in the following theorem.

**Theorem 1 (Theorem 3.1 in [1])** *Let $\xi \in \mathbb{R}^n$ be a Gaussian vector with $\mathrm{E}[\xi] = 0$ and $\mathrm{E}[\xi\xi^\top] = I_n$, $\Delta \subset \mathbb{R}^{m\times n}$, and $0 < \zeta < 1$. There exists a numerical constant $C$ such that*

$$
\begin{aligned}
&\sup_{\mathbf{Q}\in\Delta} \left| \|\mathbf{Q}\xi\|_2^2 - \mathrm{E}[\|\mathbf{Q}\xi\|_2^2] \right| \\
&\le C \left( E + V\sqrt{\log(2\zeta^{-1})} + U \log(2\zeta^{-1}) \right)
\end{aligned}
$$

*holds with probability $1 - \zeta$, where*

$$
\begin{aligned}
E &:= \gamma_2(\Delta, \|\cdot\|) \left[ \gamma_2(\Delta, \|\cdot\|) + d_{\mathrm{F}}(\Delta) \right], \\
V &:= d_{\mathrm{S}}(\Delta) \left[ \gamma_2(\Delta, \|\cdot\|) + d_{\mathrm{F}}(\Delta) \right], \\
U &:= d_{\mathrm{S}}^2(\Delta).
\end{aligned}
$$

We apply Theorem 1 to the set $\Delta = \{\mathbf{Q}_\mathbf{M} : \mathbf{M} \in \kappa(\alpha, R)\}$. The radii of $\Delta$ with respect to the Frobenius norm and to the spectral norm are respectively upper-bounded as follows:

$$d_\mathrm{F}(\Delta) \leq \alpha \sqrt{d_2}$$

and

$$d_\mathrm{S}(\Delta) \leq \frac{\alpha}{\sqrt{L}}.$$

Let $B_\mathrm{S}$ denote the unit ball with respect to the spectral norm. Then the $\gamma_2$-functional of $\Delta$ with respect to the spectral norm is upper-bounded through Dudley's inequality by

$$\gamma_2(\Delta, \|\cdot\|_2) \leq c \int_0^\infty \sqrt{\log N\left(\Delta, \eta B_\mathrm{S}\right)} \, d\eta$$

$$\leq \frac{c}{\sqrt{L}} \int_0^\infty \sqrt{\log N\left(\kappa(\alpha, R), \eta B_{1 \to 2}\right)} \, d\eta$$

$$\leq \frac{c' R \sqrt{d} \log^{3/2} d}{\sqrt{L}},$$

where the last inequality follows from Lemma 4.

Then $E$, $U$, and $V$ in Theorem 1 are upper-bounded by

$$E \leq \alpha R \sqrt{\frac{(d_1 + d_2)d_2}{L}} \log^{3/2} d$$

$$+ \frac{R^2}{L} d \log^3 d$$

$$U \leq \frac{\alpha^2}{L}$$

$$V \leq \frac{\alpha \sqrt{d_2}}{\sqrt{L}} \left( \frac{R\sqrt{d}}{L d_2} \log^{3/2} d + \alpha \right).$$

By plugging in these upper estimates to Theorem 1, we obtain

$$\sum_{\mathbf{M} \in \kappa(\alpha, R)} \left| \frac{\|\mathbf{Q}_\mathbf{M} \xi\|^2}{d_2} - \frac{\|\mathbf{M}\|_\mathrm{F}^2}{d_2} \right|$$

$$\leq c \left( \frac{\alpha R \sqrt{d} \log^{3/2} d}{\sqrt{L d_2}} + \frac{R^2 d \log^3 d}{L d_2} \right) + t$$

$$\leq \frac{c R \sqrt{d}}{\sqrt{L d_2}} \left( \alpha \log^{3/2} d + \frac{R\sqrt{d} \log^3 d}{\sqrt{L d_2}} \right) + t$$

with probability at least $1 - 2 \exp(-\hat{c} \min(t^2/V^2, t/U))$.

We take $t = \alpha R \sqrt{d} \log^{3/2} d / L d_2$ not to increase the upper bound in order. This leads to the Lemma 2.

## 3 Upper bound on $T_1$ and $T_2$

A tail bound for $T_1$ can be derived by the following lemma [2], which is a direct consequence of the moment version of Dudley's inequality (e.g., p. 263 in [3]) and a version of Markov's inequality (e.g., Proposition 7.11 in [3]).

**Lemma 1** *Let $\mu \in \mathbb{C}^n$ be a standard complex Gaussian vector with $\mathrm{E}\, \mu\mu^* = I_n$, and let $\Delta \subset \mathbb{C}^n$, $0 < \zeta < e^{1/2}$. Then, there exists constant $c$ such that*

$$\sup_{f \in \Delta} |f^* \mu| \leq c \sqrt{\log(\zeta^{-1})} \int_0^\infty \sqrt{\log N(\Delta, \|\cdot\|_2, t)} dt$$

*with probability $1 - \zeta$.*

We apply Lemma 1 to the maximum of linear forms of a Gaussian vector $\mu = [\mathbf{b}_{1,1}^\top \cdots \mathbf{b}_{L,d_2}^\top]^\top$ over the set $\mathcal{F} = \{f_\mathbf{M} : \mathbf{M} \in \kappa(\alpha, R)\}$, where $f_\mathbf{M}$ is defined by

$$f_\mathbf{M} := \left[ \mathbf{1}_{1,L} \otimes (\mathbf{M}\mathbf{e}_1)^\top \quad \ldots \quad \mathbf{1}_{1,L} \otimes (\mathbf{M}\mathbf{e}_{d_2})^\top \right]^\top.$$

Here $\mathbf{1}_{1,L}$ is the row vector of length $L$ with all entries set to 1. Then we have

$$\begin{aligned} \|f_\mathbf{M} - f_{\mathbf{M}'}\|_2 &= \|\mathbf{M} - \mathbf{M}'\|_\mathrm{F} \sqrt{L} \\ &\leq \|\mathbf{M} - \mathbf{M}'\|_{1\to 2} \sqrt{Ld_2}. \end{aligned}$$

Hence,

$$N(\mathcal{F}, \eta B_2) \leq N\left( \kappa(\alpha, R), \frac{\eta}{\sqrt{d_2}} B_\epsilon \right).$$

Combining these quantities and the entropy estimate for $N(\kappa(\alpha, R), \frac{\eta}{\sqrt{d_2}} B_\epsilon)$ with the above lemma, we get

$$\sup_{\mathbf{M}\in\kappa(\alpha,R)} \left| \sum_{l,i} \langle \mathbf{b}_{l,i}, \mathbf{M}\mathbf{e}_i \rangle \right| \leq c \log^{1/2} d \sqrt{L} R \sqrt{d}.$$

Using this, we get

$$T_1 = \mathrm{E} \, ||| \sum_{l,i} \nu_{l,i} \mathbf{A}_{l,i} |||_* \leq c\sigma \sqrt{d_2} R \sqrt{d} \log^{3/2} d$$

with probability at least $1 - 2\exp(-cd)$

Using Lemma 2, we have

$$T_2 \leq \alpha \sqrt{d_2 \left( \frac{cR}{\alpha} \sqrt{\frac{d_+ d_2}{Ld_2}} + 1 \right) \log^3 d}$$

Note that $T_1$ dominates $T_2$ when $Ld_2 < d_1 d_2$. In this case, we conclude that

$$\frac{||| \sum_{l,i} \nu_{l,i} \mathbf{A}_{l,i} |||_*}{d_2} \leq c\sigma \sqrt{L} R \sqrt{\frac{d}{Ld_2}} \log^3 d.$$

## 4 Details of the ADMM based algorithm

We now give closed form solutions to each of the update step in Algorithm 1.

### 4.0.1 Update for T

$$\begin{aligned} \mathbf{T}^{k+1} &= \arg\min_\mathbf{T} L(\mathbf{X}, \mathbf{W}^k, \mathbf{Z}^k) \\ &= \arg\min_{\mathbf{T}\succeq 0} \lambda_1 \langle [\begin{smallmatrix} \mathbf{I} & \mathbf{0} \\ \mathbf{0} & \mathbf{0} \end{smallmatrix}], \mathbf{T} \rangle + \langle \mathbf{Z}, \mathbf{T} - \mathbf{W} \rangle + \frac{\rho}{2} \|\mathbf{T} - \mathbf{W}\|_F^2 \\ &= \pi_{\mathcal{S}_+^d}(\mathbf{W}^k - \rho^{-1}(\mathbf{Z}^k + \lambda_1 [\begin{smallmatrix} \mathbf{I} & \mathbf{0} \\ \mathbf{0} & \mathbf{0} \end{smallmatrix}])) \end{aligned}$$

where $\pi$ denotes the projection operator and $\mathcal{S}_+^d$ s the set of PSD matrices of size $d$.

### 4.0.2 Update for W

$$\mathbf{W}^{k+1} = \arg\min_\mathbf{W} L(\mathbf{T}^{k+1}, \mathbf{W}, \mathbf{Z}^k)$$

This optimization can be separate into four sub-problems. Let $\mathbf{C} = \mathbf{T}^{k+1} + \rho^{-1}\mathbf{Z}^k$. Let $\widetilde{\mathbf{M}}$ be the matrix obtained by setting the diagonal elements of any matrix $\mathbf{M}$ to 0 and let $q = \mathrm{diag}(\mathbf{C}_{22})$ The four sub-problems are

1. $\mathbf{W}_{12}^{k+1} = \underset{\|\mathbf{W}_{12}\|_{1\to 2}\leq\alpha}{\arg\min} f(\mathbf{W}_{12}) + \langle\mathbf{Z}_{12}^k, \mathbf{T}_{12}^{k+1} - \mathbf{W}_{12}\rangle + \frac{\rho}{2}\left\|\mathbf{X}_{12}^{k+1} - \mathbf{W}_{12}\right\|_F^2$ where
$f(\mathbf{W}_{12}) = \sum_{l,i} |y_{l,i} - \langle A_{l,i}, \mathbf{W}_{12}\rangle|^2$

2. $\mathbf{W}_{11}^{k+1} = \arg\min_{\mathbf{W}_{11}} \|\mathbf{W}_{11} - \mathbf{C}_{11}\|_F^2$

3. $\widetilde{\mathbf{W}}_{22}^{k+1} = \arg\min_{\widetilde{\mathbf{W}}_{22}} \left\|\widetilde{\mathbf{W}}_{22} - \widetilde{\mathbf{C}}_{22}\right\|_F^2$

4. $\text{diag}(\mathbf{W}_{22}^{k+1}) = \underset{u\in\mathbb{R}^{d_2}}{\arg\min} \lambda_2 \|u\|_\infty + \frac{\rho}{2}\|u - q\|_2^2$

Sub-problem 1 is a least-squares problem which has a closed form solution. Sub-problems 2 and 3 are readily solved by setting $\mathbf{W}_{11}^{k+1} = \mathbf{C}_{11}$ and $\widetilde{\mathbf{W}}_{22}^{k+1} = \widetilde{\mathbf{C}}_{22}$. Sub-problem 4 has a closed form solution as described in [4].

# 5 Entropy Estimates of Tensor Products

For symmetric convex bodies $D$ and $E$, the *covering number* $N(D, E)$ and the *packing number* $M(D, E)$ are respectively defined by

$$N(D, E) := \min\left\{l \,\middle|\, \exists y_1, \ldots, y_l \in D, \, D \subset \bigcup_{1\leq j\leq l} (y_j + E)\right\},$$

$$M(D, E) := \max\left\{l \,\middle|\, \exists y_1, \ldots, y_l \in D, \, y_j - y_k \notin E, \, \forall j \neq k\right\}.$$

Let $X, Y$ be Banach spaces. For $T \in L(X, Y)$, the *dyadic entropy number* [5] is defined by

$$e_k(T) := \inf\{\epsilon > 0 \,|\, M(T(B_X), \epsilon B_Y) \leq 2^{k-1}\}.$$

where $B_X$ and $B_Y$ denote unit balls. We will use the following shorthand notation for the weighted summation of the dyadic entropy numbers:

$$\mathcal{E}_{2,1}(T) := \sum_{k=1}^{\infty} k^{-1/2} e_k(T),$$

which is up to a constant equivalent to the entropy integral $\int_0^\infty \sqrt{\ln N(T(B_X), \epsilon B_Y)}d\epsilon$ [6], which plays a key role in analyzing properties on random linear operators on low-rank matrices.

In this section, we derive the $\mathcal{E}_{2,1}$ of the identity operator from the injective tensor product to the projective tensor product of a set of Banach space pairs. Note that these tensor product spaces are valid Banach spaces too. The main machinery in deriving these estimates is Maurey's empirical method [7], summarized in the following lemma.

**Lemma 2** *Let $T \in \ell_\infty^{d_2} \otimes \ell_\infty^{d_1}$. Then*

$$\mathcal{E}_{2,1}(T) \leq C\sqrt{1 + \ln(d_1 \vee d_2)}\,(1 + \ln(d_1 \wedge d_2))^{3/2}\|T\|_\vee.$$

To apply Lemma 2 to $\ell_\infty^{d_2} \otimes \ell_p^{d_1}$ with $2 \leq p < \infty$, we need the following result that shows embedding of finite dimensional $\ell_p$ space to $\ell_1$ up to a small Banach-Mazur distance.

**Lemma 3 ([7, Lemma 5])** *Let $1 < p \leq 2$ and $\epsilon > 0$. There is a constant $c(p, \epsilon) > 0$ for which the following property holds: For each $d_1$, there exists $k \geq c(p, \epsilon)d_1$ so that $\ell_1^{d_1}$ contains a subspace $(1 + \epsilon)$-isomorphic to $\ell_p^k$, i.e., the Banach-Mazur distance is upper-bounded by $(1 + \epsilon)$.*

Then we obtain the following entropy estimate for $\ell_\infty^{d_2} \otimes \ell_p^{d_1}$ with $2 \leq p < \infty$ by combining Lemmas 2 and 3.

Let $2 \leq p < \infty$. Then

$$\mathcal{E}_{2,1}(\text{id} : \ell_\infty^{d_2} \widehat{\otimes} \ell_p^{d_1} \to \ell_\infty^{d_2} \otimes \ell_p^{d_1}) \leq C\sqrt{d_1 + d_2}\,(1 + \ln(d_1 d_2))^{3/2}.$$

Note that Lemma 4 in the main paper is a particular case of Lemma 3 above.