[Reviews · NeurIPS 2019]

Reviewer 1



I have a few major questions about the paper that should be addressed in a published version of this paper. * Does the sensing model interact with the new choice of norm? * Compare to the literature on matrix sensing as formulated in [1]. Why do recovery results for matrix sensing not apply to this problem setting? The sensing matrices A would have only one nonzero column each... * Why do you call this model "decentralized"? Recovery still requires localizing all the measurements to a single master node (to solve the SDP). And it is easy to implement more general matrix sensing so it is equally decentralized: just add up the measurements from each column to form the measurement of the full matrix. (Perhaps the matrix sensing approach to decentralized sketching requires more measurements?) * Does the mixed norm have an interpretation as a spectral function, or is it not equivalent to any function of the spectrum of the matrix? * The experiments should be improved. In particular, the authors should also compare with max norm and other more exotic norms such as [2] (or explain why they are inferior). How were the parameters chosen, such as the radius R? * Please say more about what elements of your proof are standard, and which elements are new to adapt to either the sensing structure or the mixed norm. and a few detailed comments: * line 135: consider citing [3], which explores tensors norms of this form in detail in the context of matrix estimation problems. * line 139: I found the notation for injective vs projective norms distracting. Is it standard? [1] Recht, Benjamin, Maryam Fazel, and Pablo A. Parrilo. "Guaranteed minimum-rank solutions of linear matrix equations via nuclear norm minimization." SIAM review 52.3 (2010): 471-501. [2] Shang, Fanhua, Yuanyuan Liu, and James Cheng. "Tractable and scalable Schatten quasi-norm approximations for rank minimization." Artificial Intelligence and Statistics. 2016. [3] John Bruer. "Recovering Structured Low-rank Operators Using Nuclear Norms." PhD Thesis, Caltech, 2018. https://thesis.library.caltech.edu/10048/

Reviewer 2



The critical part of the algorithm/analysis seems to the notation of the mixed norm defined in Eq. (4) and so the traditional recovery approach can be adapted to a carefully defined set of data matrices depending on the max ell_2 column norm and the mixed norm. The algorithm is just to solve a least square minimization over that set of data matrices. Is it true that the measurement matrices need not be of Gaussian entries and can be any symmetric subgaussian entries? A key entropy estimate is Lemma 2, which lacks a proof, not even in the supplementary. The paper seems to be written in a rush and is clearly not proofread. It is infested with writing problems, particularly seriously concerning the definition, and thus impedes understanding. The following is not an exhaustive list: Line 48: say ‘Assume’ instead of ‘It follows’ Equation (2): what is the inner product? Equation (3): the subscript field of ‘max’ on the right-hand side should be j = 1,\dots,d_2 Equation (4): What are the dimensions of U and V? Arbitrary matching dimensions? Line 131: maybe just say X is a map from (R^{d_2}, ell_p) to (R^{d_1}, ell_q), which is a cleaner expression. Equation (9): What is the use of u? Is it ||Xu||_q? Is the dimension of u correct? Line 143: what is the *-norm? Trace norm? Line 144: What is the infinity-norm? Maximum absolute column sum?

Reviewer 3



The paper is very clearly written. I particularly appreciated the simplicity of the exposition both in the main paper and the appendix. The quality of the presentation is high. The authors clearly outline their techniques at multiple levels, and also discuss possible obstacles along the way and how they resolved them. The techniques seem original. The main component seems to be the use of the new mixed norm as a regularizer in low-rank matrix recovery. This is not altogether surprising, since earlier papers by Lee, Srebro, and co-authors also proposed similar norms in the same context. However, such an approach to the compressive PCA setting is to my knowledge not very common, and gives improved (for-all) guarantees in contrast with earlier work by Negahban and Wainwright. Compressive PCA is a somewhat more challenging problem than regular low-rank matrix recovery from affine measurements, because of the tensor-product structure of the linear observation model. To my knowledge, most existing theoretical results for low-rank matrix recovery do not quite cover this setting. The results of this paper fill this gap. === Update (after reading rebuttal/reviews): thanks for the response. Consider adding real-data experiments, as well as incorporating other suggestions by the reviewers.

[Author Response · NeurIPS 2019]

We thank the reviewers and the ACs for their insightful comments. Please find our point-to-point responses below.

**Reviewer #2**: We find the reviewer's suggestions very helpful in improving the quality of the manuscript. We will address the various points raised and include this discussion in the updated version of the paper. - Our choice of using the mixed-norm was motivated by the sensing model. Each sensing matrix is the outer product of a Gaussian vector and a standard basis vector. The former has its energy spread across all coordinates whereas the latter is localized. We chose Banach spaces for the domain and range accordingly and we chose the appropriate tensor norms, to build a unifying view on previous relevant results. On the technical front, the measurements obtained under our sensing model (Equation 2) form an embedding of the set $\kappa(\alpha, R)$, resulting in an optimal sample complexity. The proof depends on the entropy number of $\kappa(\alpha, R)$ with respect to the maximum-column norm, which has a favorable dependence on the number of degrees of freedom (Lemma 4). Hence, using the mixed-norm as a characterization of low-rank matrices is critical for our sensing model. - The guarantee on nuclear norm minimization by Recht et al. was based on the restricted isometry property (RIP). However relevant negative results on the RIP of rank-one measurements have been shown [7]. Instead, we compared our results to a more recent approach with the nuclear norm [11] in the manuscript. - While averaging measurements over columns provides a measurement with a full Gaussian matrix, the resulting number of measurements is smaller by a factor of the number of columns. Therefore the sample complexity increases accordingly. - We agree that the usage of the term "decentralized" is not consistent with that in the optimization literature and it might cause confusion among the readers. Therefore we will replace "decentralized sketching" by "distributed sketching". - In general, tensor-product norms are not given as a function of spectrum of a matrix. One exception is when the domain and range are Hilbert spaces. - With regards to comparison with max-norm regularization, since our choice of the tensor norm is according to the structure of the measurement procedure, such a comparison will help illustrate the importance of model-based design of our convex program. It would also be interesting to see how the method competes with the quasi-norm-based method by Shang et al. along with other nonconvex optimization approaches. We plan to demonstrate these experimental results in an extended journal version where we also plan to present the generalization of the theory to a broader class of tensor norms. - Although the parameter $R$ is not known a priori, it is possible to tune it via the following heuristic: one can start with low values of $R$, resulting in higher residuals (since the ground truth matrix is not in the set) and increase $R$ until the residual plateaus. - Our proof up to (16) was inspired by the analogous part in [7] but deviates significantly after this. We have derived the concentration bound on the quadratic term in (16) by specializing the suprema of second order chaos processes to our sensing model. More importantly, the entropy estimate to bound the Talagrand $\gamma_2$-functional is new and has been derived based on our own extension of Maurey's empirical lemma [reference 7, supplementary material] from $\ell_1^n$ to a set of Banach spaces. - We found that Bruer's PhD thesis is highly relevant and explored many inspiring algorithms and experiments. We will add a discussion comparing this thesis to our work in the updated version of our paper. - Our notation for the injective and projective tensor norms has been borrowed from [21].

**Reviewer #3**: Thank you for the detailed comments on notations and typos. Although some notation was adapted using conventions in the literature, we will further clarify these in the context of our paper for better readability. The inner product in eqn (2) is given as $\langle \boldsymbol{A}, \boldsymbol{B} \rangle = \mathrm{trace}(\boldsymbol{B}^\top \boldsymbol{A})$. The norm with subscript $*$ denotes the nuclear norm, that is, the sum of all singular values, and coincides with the trace of the matrix when it is positive semidefinite. The norm with subscript $\infty$ denotes the $\ell_\infty$-norm, that is, the maximum absolute entry. We will also make the following edits as per the reviewer's suggestions: The mixed-norm is defined as the minimum over all possible factorizations of the matrix $\mathbf{X}$, the product of the Frobenius norm of the left factor and the maximum column norm of the right factor and in Eqn (4), $\mathbf{U}$ and $\mathbf{V}$ can have any arbitrary matching dimensions; Line (48) should say 'Assume'; Eqn (3) considers the maximun over the indices $j = 1, \cdots, d_2$; The right-hand side of eqn (9) should be $\sup_{\mathbf{u} \in \mathbb{R}^{d_2}, \|\mathbf{u}\|_p = 1} \|\mathbf{X}\mathbf{u}\|_q$. The reviewer also suggested to consider the extension to subgaussian case. - We have verified that our theoretical result remains valid when the non-zero entries of the sensing matrices are drawn from any symmetric subgaussian distribution. - As for the entropy estimate in Lemma 2, we did not include the full proof to the supplementary material because we plan to present the result and its generalization to other pairs of tensor norms in a separate journal submission. We have hence included only a sketch of the proof.

**Reviewer #4**: We appreciate the reviewer for many constructive suggestions. We aim to conduct experiments on large-scale datasets from various real-world applications including hyperspectral imaging (AVIRIS dataset), fMRI (www.humanconnectome.org), neural recordings, and the MovieLens datset. Data in these domains have high dimensions owing to the measurement resolution, but it is common for them to have low dimensional structure. - As the reviewer suggested, extending our algorithm/analysis to handle outliers could be interesting for various applications. For example, in genomics and video data, outliers are frequently observed and add serious artifacts to the analysis. Additionally we will further extend our findings to a broader class of tensor-norm models with applications including those studied in Bruer's thesis (suggested by Reviewer #2). - In order to compare our methods against max-norm and others suggested by reviewer #2 and in order to experiment on real datasets, we need to study further the computational aspects of our algorithm. In particular, we plan to focus on how to tune the parameters $(\alpha, R)$, how to parallelize the computation and also explore different techniques to choose the step-size in the update steps.

[Meta-Review · NeurIPS 2019]

Overall the reviewers appreciated the theoretical results and thought this filled in a natural hole in the literature. Some concerns about the writing (though their were differing views on the quality of this) and experiments were raised, though these were minor compared to the theoretical part, thus lifting this paper over the bar.